# Occupational Heat Stress: Multi-Country Observations and Interventions

**DOI:** 10.3390/ijerph18126303

**Published:** 2021-06-10

**Authors:** Leonidas G. Ioannou, Konstantinos Mantzios, Lydia Tsoutsoubi, Eleni Nintou, Maria Vliora, Paraskevi Gkiata, Constantinos N. Dallas, Giorgos Gkikas, Gerasimos Agaliotis, Kostas Sfakianakis, Areti K. Kapnia, Davide J. Testa, Tânia Amorim, Petros C. Dinas, Tiago S. Mayor, Chuansi Gao, Lars Nybo, Andreas D. Flouris

**Affiliations:** 1FAME Laboratory, Department of Physical Education and Sport Science, University of Thessaly, 42100 Trikala, Greece; ioannouLG@gmail.com (L.G.I.); konstantinosmantzios@gmail.com (K.M.); lydiatsoutsoubi@gmail.com (L.T.); enintou@gmail.com (E.N.); mvliora@gmail.com (M.V.); gkiata.vivi@gmail.com (P.G.); constantinedallas@gmail.com (C.N.D.); ggkikas@uth.gr (G.G.); gagaliotis@uth.gr (G.A.); ksfakianakis98@gmail.com (K.S.); Areti.kapnia@gmail.com (A.K.K.); davide.testa13@gmail.com (D.J.T.); tania.patricia.amorim@gmail.com (T.A.); petros.cd@gmail.com (P.C.D.); 2Department of Nutrition, Exercise and Sports, August Krogh Building, University of Copenhagen, 2100 Copenhagen, Denmark; nybo@nexs.ku.dk; 3SIMTECH Laboratory, Transport Phenomena Research Centre, Engineering Faculty of Porto University, 4200-465 Porto, Portugal; tiago.sottomayor@fe.up.pt; 4Thermal Environment Laboratory, Division of Ergonomics and Aerosol Technology, Department of Design Sciences, Faculty of Engineering, Lund University, 22100 Lund, Sweden; chuansi.gao@design.lth.se

**Keywords:** heat stress, work, mitigation, labor productivity, physiological strain, hydration, breaks, ventilated garments, mechanization, ice slurry

## Abstract

Background: Occupational heat exposure can provoke health problems that increase the risk of certain diseases and affect workers’ ability to maintain healthy and productive lives. This study investigates the effects of occupational heat stress on workers’ physiological strain and labor productivity, as well as examining multiple interventions to mitigate the problem. Methods: We monitored 518 full work-shifts obtained from 238 experienced and acclimatized individuals who work in key industrial sectors located in Cyprus, Greece, Qatar, and Spain. Continuous core body temperature, mean skin temperature, heart rate, and labor productivity were collected from the beginning to the end of all work-shifts. Results: In workplaces where self-pacing is not feasible or very limited, we found that occupational heat stress is associated with the heat strain experienced by workers. Strategies focusing on hydration, work-rest cycles, and ventilated clothing were able to mitigate the physiological heat strain experienced by workers. Increasing mechanization enhanced labor productivity without increasing workers’ physiological strain. Conclusions: Empowering laborers to self-pace is the basis of heat mitigation, while tailored strategies focusing on hydration, work-rest cycles, ventilated garments, and mechanization can further reduce the physiological heat strain experienced by workers under certain conditions.

## 1. Introduction

Occupational heat stress (OH-stress) increases the physiological heat strain experienced by workers (OH-strain) leading to a higher risk of diseases and health problems including dehydration [1], occupational accidents [2,3,4], absenteeism [5], as well as chronic kidney injuries [6]. Moreover, OH-strain leads to reduced capacity for work and lower productivity output [6,7,8]. By the end of the century, five out of ten people will be exposed to harmful climatic conditions under a mild climate change scenario (assuming drastic reduction of greenhouse gases) and more than seven out of ten people under a scenario of growing gas emissions [9]. This, of course, will be more pronounced for workers in agriculture and construction [10], due to the requirement to perform manual labor outdoors and the lack of cost-effective heat mitigation strategies for these occupational settings. Beyond these obvious cases of occupational heat exposure, climate-dependent industrial sectors such as tourism are likely to experience a lengthening of their summer seasons [11], leading their workers to experience a more prolonged working season in the heat.

Working in climate-vulnerable industries (e.g., agriculture, construction, and tourism) often translates to prolonged manual labor in hot environments. While it would be wrong to assume that all workers in these industries are exposed to OH-stress, this issue certainly involves a very large number of workers since these sectors currently employ about half of the world’s labor force: agriculture employs ~32% [12]; construction employs ~10% [13]; and tourism employs ~8% [14] of all workers. The monetary turnover associated with these sectors plays a primary role in the global economy, while the projected heat-induced labor loss in monetary terms is expected to be as high as 2216 billion EUR (2400 billion USD) by 2030 [10,15].

Protecting workers’ health while maintaining labor productivity is vital [16,17,18] and has driven many researchers to investigate the effectiveness of different heat mitigation strategies [19,20]. Specifically, heat mitigation strategies such as planned breaks [21], use of personal cooling vests [22,23], ice slurry consumption [22,24], optimized clothing [25], and ventilated garments [26,27] were previously tested in lab settings for mitigating the OH-strain experienced by individuals performing physical work (typically exercise). However, there is limited knowledge regarding the capacity of these heat mitigation strategies to protect workers’ health and labor productivity in occupational settings, taking into account real-life work-related parameters. For example, job task requirements and complexities, self-pacing, and sun exposure are only some of the parameters that are difficult to consider in laboratory settings and are known to affect the physiological heat strain experienced by someone [7,28]. To address this knowledge gap, a series of observational and interventional field studies were conducted in different countries around the globe. The first aim of the current study was to investigate the effects of OH-stress on the thermo-physiological responses and labor productivity of workers in agriculture, construction, and tourism sectors in different countries. The second aim of the study was to examine different cost-effective heat mitigation strategies to reduce the OH-strain experienced by workers in the aforementioned industrial sectors, without jeopardizing their capacity to perform labor.

## 2. Materials and Methods

The experimental protocol (ClinicalTrials.gov ID: NCT04160728) for these field experiments was approved by the Bioethical Committee of Department of Physical Education and Sport Science of the University of Thessaly and the National Bioethical Review Board of Cyprus in accordance with the Declaration of Helsinki. This series of field experiments involved monitoring typical acclimatized adult workers who live in the area and perform similar work tasks on a daily basis. No exclusion criteria, other than not being an adult, experienced, and acclimatized worker, were applied. Therefore, workers with different sex, age, body mass, body stature, ethnicity, education, and socio-economic background were recruited in the present study.

### 2.1. Observational Studies

To examine the effects of OH-stress on the thermo-physiological responses and labor productivity of workers in agriculture, construction, and tourism sectors, we collected physiological (core temperature, mean skin temperature, and heart rate) and labor (work intensity) data from 99 workers during full work-shifts in agriculture (Greece), construction (Spain), and tourism (Greece) industries. Prior to their participation in the study, written informed consent was obtained from all volunteers after a detailed explanation of all the procedures involved. It is important to note that in our measures in the agriculture sector of Greece, there were incorporated workers (Romani ethnic group) who did not consent to measurements of core temperature, mean skin temperature, and heart rate as well as to assessing their personal characteristics (body mass, body stature, and age). Nevertheless, they agreed to participate in the study and have their labor data (see below “time-motion analysis”) collected.

Self-reported age, body stature, and body mass were collected prior to the study. Continuous core temperature (T_core_) and mean skin temperature (T_sk_) data were collected using telemetric capsules (BodyCap, Caen, France), and wireless thermistors (iButtons type DS1921H, Maxim/Dallas Semiconductor Corp., USA), respectively. Skin temperature data were collected from four body sites (chest, arm, thigh, and leg) and were expressed as T_sk_ using the formula of Ramanathan (T_sk_ = 0.3(chest + arm) + 0.2(thigh + leg)) [29]. Furthermore, continuous environmental data (air temperature (°C), globe temperature (°C), relative humidity (%), and air velocity (m/s)) were collected using portable weather stations (Kestrel 5400FW, Nielsen-Kellerman, Pennsylvania, USA) installed in close proximity to the workers. The same weather station provides measures of Wet-Bulb Globe Temperature (WBGT) with an accuracy of ±0.7 °C in ambient temperatures ranging between −29 and 70 °C according to the developer’s manual. An established method of second-by-second time motion analysis was utilized to examine the labor effort of agriculture workers [7,30], while real-time task analysis was utilized to evaluate the labor intensity of workers in tourism and construction sectors. Real-time task analysis is based on the premise of the aforementioned time-motion analysis. The only difference between the two methods is that real-time task analysis is performed live at the work site, while time-motion analysis is based on video recordings taken at the work site and analyzed at a later time. It is important to note that the data collection took place across multiple days characterized by different ambient conditions. All the data collected throughout the experiments are presented in Figure 1.

Time-motion analysis was conducted for 11 different activities. Previous knowledge related to the energy cost of different activities [31,32] was used to determine the intensity characterizing each work task (Table A1). Real-time task analysis was conducted using different levels of work intensity (i.e., rest, low/medium/high intensity) based on the International Standard 8996 on the determination of metabolic rate. Specifically, rest (65 W/m^2^) was characterized as any activity involving resting and/or sitting at ease. Low-intensity labor (100 W/m^2^) included activities incorporating “hand and arm work” or “hand and leg work”, such as driving vehicles in normal conditions, machining, and casual walking (at a speed up to 2.5 km/h). Moderate-intensity labor (165 W/m^2^) included activities involving “hand and arm work”, “arm and leg work”, or “arm and trunk work” such as working with construction equipment, weeding, picking fruits, or walking (at a speed of 2.5–5.5 km/h). High-intensity labor (230 W/m^2^) included any activity involving using intense arm and trunk work, carrying heavy material, pushing or pulling heavily, or walking (at a speed of 5.5–7.0 km/h).

### 2.2. Interventional Studies

To examine the capacity of different heat mitigation strategies to reduce the OH-strain experienced by workers across various industrial sectors, we compared a “business as usual” (BAU) scenario, where workers followed their normal work routine, against different heat mitigation strategies in each industrial sector across different countries (Cyprus, Greece, Qatar, and Spain). The BAU and heat mitigation strategies were randomly allocated among four full work-shifts and were tested after consulting with the management of each worksite, aiming to test economically viable and feasible interventions without jeopardizing workers’ normal workflow. All the tested strategies incorporated the same data collection (Figure 1) and procedures as during the observational field experiments. In addition, continuous heart rate (HR) was collected using wireless heart rate monitors (Polar Team2, Polar Electro Oy, Kempele, Finland), as well as urine samples collected at the start and the end of the BAU and hydration scenarios to evaluate the hydration status of each worker. Urine-specific gravity was assessed for each urine sample using a refractometer (PAL-10S, ATAGO CO., LTD., Fukaya, Saitama Prefecture, Japan) and was classified as either hydrated (<1.020) or dehydrated (≥1.020) [33].

#### 2.2.1. Agriculture Industry

In agriculture, we tested three different strategies in Cyprus (06:00–14:00 work-shift) and three more strategies in Qatar (04:30–11:00 work-shift). In Cyprus, agriculture workers were provided with (1) “planned breaks” (i.e., 90 s of break every half hour of work) and (2) a “mechanical fruit cart” (i.e., machinery which carries up to 225 kg of crops with minimal labor effort) aiming to reduce workers’ metabolic heat production, as well as with (3) “ventilated garments” (i.e., short-sleeved shirts with integrated electric fans) aiming to increase their evaporative and convective heat losses. In Qatar, agriculture workers were (1) advised (but not forced) to drink 750 mL of water every hour (supplemented with a total of one tablespoon of salt, for the entire work-shift, to avoid hyponatremia) from the start until the end of the work-shift, (2) provided with “evaporative garments” (i.e., breathable lightweight work shirt and trousers that allowed the wearer to wet the collar, cuffs, and pocket pouches with water), and (3) provided with ten minutes of a planned break (workers were advised to rest and hydrate in the shade) every 50 min of continuous work from 06:00 to 11:00. Salt supplementation during the hydration strategy was considered only for workers who had not been diagnosed with high blood pressure and/or other cardiovascular abnormalities. Moreover, the suggested amount of salt (~15 g) was calculated based on our predictions which indicate that a typical worker is expected to secrete ~6 L of sweat during an 8-h shift in the heat [30] and the known concentration of salt in sweat (2.1 to 3.2 g/L) [34]. One day prior to the evaporative garments strategy in Qatar, workers were provided with a demonstration explaining in detail the different properties of the evaporative garments as well as how they should be used. Thereafter, the workers were free to use the evaporative garments as desired in order to assess how they would be eventually used in real-life conditions.

#### 2.2.2. Construction Industry

In construction, we tested three different heat mitigation strategies in Qatar (three different scenarios for each one of the following work-shifts: (a) 00:00–11:00, (b) 15:30–02:30, and (c) 06:00–17:00) and three more strategies in Spain (09:00–19:00 work-shift). In Qatar, we tested the (1) hydration, (2) evaporative garments, and (3) work-rest cycle strategies as described above for the agriculture workers in the same country. However, during the BAU scenario of the construction workers studied in Qatar, a raft of other heat mitigation strategies were already in place at the work site including (1) shaded areas every 100–200 m, (2) water stations every 300–400 m, (3) an obligation for each worker to carry a 1 L water bottle throughout the work-shift, (4) air-conditioned rest areas to be used during planned breaks, and (5) education on the effects of OH-stress and dehydration by a large team of safety and welfare officers, with reminders using large signs throughout the site in languages understood by workers.

In Spain, we provided construction workers with (1) 750 mL of cold water (but they were not forced to drink) every hour (supplemented with a total of one tablespoon of salt, for the entire work-shift, to avoid hyponatremia), (2) two breaks of seven minutes (at 12:30 and 16:30) in the “planned break” scenario, and (3) 300 mL of crushed ice (ice slurry) every hour. During the hydration strategy, salt supplementation was considered only for workers who had not been diagnosed with high blood pressure and/or other cardiovascular abnormalities. Moreover, the suggested amount of salt (~15 g) was calculated based on our predictions which indicate that a typical worker is expected to secrete ~6 L of sweat during an 8-h shift in the heat [30] and the known concentration of salt in sweat (2.1 to 3.2 g/L) [34]. Furthermore, during the BAU scenario of the construction workers examined in Spain, a raft of other heat mitigation strategies was already in place at the work site including (1) shaded areas every 50–100 m, (2) water stations every 50–100 m, and (3) air-conditioned rest areas to be used during planned breaks.

#### 2.2.3. Tourism Industry

Three different heat mitigation strategies were tested in the tourism industry of Greece. Specifically, tourism workers were provided with (1) 90 s of a planned break every 30 min of continuous work, (2) ice slurries (3.5 mL per body mass kilogram) every hour of continuous work, and (3) two minutes of a planned break combined with ice slurry consumption (2.4 g per body mass kilogram) every hour of continuous work.

### 2.3. Data Analysis

In observational studies, the collected WBGT data were rounded to the closest integer. Thereafter, we calculated the average T_core_, T_sk_, and metabolic rate of workers during exposure in each degree WBGT. Pearson’s correlation coefficient was used to examine the relationships between the average T_core_, T_sk_, and metabolic rate in each degree WBGT and the thermal stress (i.e., rounded WBGT values) experienced by workers. The magnitude of the observed relationships was determined based on previous literature [35]. Linear and non-linear regression analyses with prediction intervals were used to examine if thermal stress (rounded WBGT values) can predict the OH-strain (i.e., average T_core_ and T_sk_ at each degree WBGT) and labor effort (average metabolic rate at each degree WBGT) experienced by workers in each industrial sector.

To compare the different heat mitigation strategies conducted in each industrial sector against the BAU scenario, workers should be exposed to environments characterized by the same heat stress. Although all heat mitigation strategies were scheduled based on weather forecasts to imply the same thermal stress as during the BAU, in three out of the fifteen scenarios, we identified a significant difference (≥1 °C WBGT) in the thermal stress between the BAU scenario and the heat mitigation strategies. In such cases, the Predicted Heat Strain model [30] was utilized to homogenize the data by computing the expected difference in T_core_, T_sk_, and HR (33 bpm (average thermal cardiac reactivity) per degree of difference in T_core_) between the conflicted scenarios, and thereafter subtracting/adding them from/to the actual values collected in the field. For instance, in a hypothetical case where an agriculture worker (height = 180 cm and weight = 75 kg) was exposed to an environment of 29 °C WBGT (air temperature = 35 °C; globe temperature = 39.8 °C; relative humidity = 40%; and wind speed = 0.5 m/s) during the BAU scenario, compared to an environment of 30 °C WBGT (air temperature = 35.9 °C; globe temperature = 41.0 °C; relative humidity = 41%; and wind speed = 0.6 m/s) during the hydration strategy, we computed the expected mean differences in T_core_, T_sk_, and HR, and we subtracted them from the actual mean values during the hydration scenario. In this case, 0.12 °C, 0.19 °C, and 4.11 bpm were subtracted from the actual collected T_core_, T_sk_, and HR, respectively. Thereafter, paired t-tests alongside Cohen’s d effect sizes were used to examine possible differences between the BAU scenario and all the tested heat mitigation strategies. The magnitude of effect sizes [36] was determined as follows: d (0.01) = very small; d (0.2) = small; d (0.5) = medium; d (0.8) = large; d (1.2) = very large; and d (2.0) = huge. Relative risks were calculated to investigate the odds of being dehydrated (i.e., prevalence of dehydration) during the BAU and hydration scenarios. The relative risks, their standard error, and 95% confidence intervals were calculated based on previous methodology [37].

Statistical analyses were conducted using the SPSS v25.0 (IBM, Armonk, NY, USA), the OriginPro 2020 (OriginLab Northhampton, MA, USA) and the Excel spreadsheets (Microsoft Office, Microsoft, Washington, USA). The level of significance for these analyses was set at *p* < 0.05.

## 3. Results

The study involved monitoring 518 full work-shifts obtained from 238 experienced and acclimatized (living in the area for more than two months) workers. Detailed information on participants’ anthropometric characteristics is presented in Table 1.

### 3.1. Observational Studies

To investigate the effects of OH-stress on the thermophysiological responses and labor effort of workers in agriculture, construction, and tourism sectors, 99 workers from Cyprus, Greece, and Spain were monitored during their full work-shift. The environmental conditions during the observations in agriculture (air temperature: 23.1 ± 6.4 °C; globe temperature: 35.0 ± 6.6 °C; relative humidity: 50.4 ± 8.8%; and air velocity: 1.2 ± 0.8 m/s) ranged from 14.5 to 30.3 °C WBGT. The environmental conditions during the observations in construction (air temperature: 26.6 ± 3.9 °C; globe temperature: 32.1 ± 8.3 °C; relative humidity: 49.8 ± 13.3%; and air velocity: 0.4 ± 0.8 m/s) ranged from 19.2 to 29.2 °C WBGT. The environmental conditions during the observations in the tourism industry (air temperature: 29.8 ± 2.6 °C; globe temperature: 31.1 ± 3.7 °C; relative humidity: 54.3 ± 8.5%; and air velocity: 0.2 ± 0.4 m/s) ranged from 20.2 to 32.4 °C WBGT.

Strong relationships were identified between WBGT and the T_sk_ of people who worked in agriculture (r = 0.970, *p* < 0.001), construction (r = 0.922, *p* < 0.001), and tourism (r = 0.595, *p* = 0.032) sectors (Figure 2). Similarly, linear regressions demonstrated that there is a 0.31 °C, 0.23 °C, and 0.09 °C increase in T_sk_ for every 1 °C increase in WBGT for people who work in agriculture (R^2^ = 0.941; F_(1,10)_ = 159.098, *p* < 0.001), construction (R^2^ = 0.850; F_(1,8)_ = 45.443, *p* < 0.001), and tourism (R^2^ = 0.354; F_(1,11)_ = 6.029, *p* = 0.032) sectors, respectively (Figure 2 and Table A2).

No relationship was identified between WBGT and the T_core_ of agriculture workers (r = −0.052, *p* = 0.872). On the other hand, strong relationships were identified between WBGT and the T_core_ of people who work in construction (r = 0.765, *p* = 0.010) and tourism (r = 0.646, *p* = 0.017) sectors (Figure 3). Furthermore, linear regressions showed no effect of OH-stress on the T_core_ of agriculture workers, while there is a 0.05 °C increase in T_core_ for every 1 °C increase in WBGT for people who work in construction (R^2^ = 0.585; F_(1,8)_ = 11.254, *p* = 0.010; Table A2). A biphasic regression (R^2^ = 0.852; F_(2,10)_ = 28.678, *p* < 0.001) demonstrated that there is an increase of ~0.4 °C in the T_core_ of tourism workers for every 1 °C increase in WBGT above 30 °C (Figure 3 and Table A2).

A very strong relationship between WBGT and the metabolic rate of agriculture workers (r = −0.787, *p* < 0.001) was found (Figure 4). On the other hand, no statistically significant relationships were found between WBGT and the metabolic rate of construction (r = −0.249, *p* = 0.487) or tourism (r = 0.035, *p* = 0.908) workers (Figure 4). Linear regressions demonstrated that there is a 3.1 W/m^2^ decrease in the metabolic rate of agriculture workers (R^2^ = 0.619; F_(1,14)_ = 22.719, *p* < 0.001) for every 1 °C increase in WBGT, but no statistically significant relationships in construction (*p* = 0.487) or tourism (*p* = 0.908) sectors (Figure 4 and Table A2).

### 3.2. Interventional Studies

To examine possible cost-effective heat mitigation strategies to reduce the OH-strain, 139 workers were monitored in agriculture, construction, and tourism industries over four full work-shifts characterized by similar thermal stress (Table A3). Overall, ten different heat mitigation strategies (five of which were investigated in different industries) were tested to examine their capacity to mitigate the OH-strain experienced by people who work under heat stress. Detailed information regarding the efficacy of each heat mitigation strategy is presented below.

#### 3.2.1. Agriculture Industry

In Cyprus, the use of the “mechanical fruit cart” did not significantly lower the T_core_ (*p* = 0.364), T_sk_ (*p* = 0.890), HR (*p* = 0.216), or labor effort (*p* = 0.308) of agriculture workers (Figure 5). However, workers picked 63% more crop when using the “mechanical fruit cart” (4400 kg) compared to the BAU scenario (2700 kg). Providing agriculture workers with “planned breaks” did not significantly alter their T_core_ (*p* = 0.492), T_sk_ (*p* = 0.893), HR (*p* = 0.079), or labor effort (*p* = 0.357) (Figure 5). On the other hand, we identified that “ventilated garments” were able to reduce the T_sk_ (*p* = 0.009; d = −1.59) of agriculture workers without impacting their T_core_ (*p* = 0.124), HR (*p* = 0.918), or labor effort (*p* = 0.483) (Figure 5) during the work-shift.

In Qatar, the hydration strategy increased workers’ HR (*p* = 0.009; d = 0.69) without affecting their T_core_ (*p* = 0.457), T_sk_ (*p* = 0.986), or labour effort (*p* = 0.779) (Figure 6). The hydration strategy also decreased the prevalence of dehydration by 54% (95% confidence interval: 1.06 to 2.24). The evaporative garments strategy did not alter the T_core_ (*p* = 0.512), T_sk_ (*p* = 0.606), HR (*p* = 0.262), or labor effort (*p* = 0.627) of agriculture workers, whereas providing them with “planned breaks” increased their HR (*p* = 0.006; d = 0.84) (Figure 6).

#### 3.2.2. Construction Industry

Heat mitigation strategies tested in Qatar showed that hydration reduced workers’ T_core_ (*p* = 0.035; d = −0.45) without affecting their T_sk_ (*p* = 0.440), HR (*p* = 0.708), or labor effort (*p* = 0.944) (Figure 7). Moreover, the hydration strategy decreased the prevalence of dehydration by 97% (95% confidence interval: 0.62 to 6.27). The evaporative garments strategy did not impact the T_core_ (*p* = 0.250), T_sk_ (*p* = 0.440), HR (*p* = 0.164), or labor effort (*p* = 0.077) of the tested construction workers (Figure 7). A point of note here was that our research team observed that many workers did not follow the manufacturer’s usage instructions for frequent wetting and often wore t-shirts underneath the evaporative garments. Providing construction workers with “planned breaks” significantly reduced their T_sk_ (*p* < 0.001; d = −0.64), HR (*p* < 0.001; d = −0.28), and labor effort (*p* < 0.001; d = −0.63) without affecting their T_core_ (*p* = 0.901) (Figure 7).

In Spain, the tested hydration strategy reduced the T_core_ (*p* = 0.022; d = −1.07) and HR (*p* = 0.009; d = −1.51) and increased the labor effort (*p* = 0.029; d = 0.39) of the tested construction workers (Figure 8). Moreover, hydration decreased the prevalence of dehydration by 13% (95% confidence interval: 0.78 to 1.63). Surprisingly, the T_sk_ was increased during this heat mitigation strategy (*p* = 0.012; d = 1.28) compared to BAU (Figure 8). The “planned breaks” strategy did not impair workers’ T_core_ (*p* = 0.492), T_sk_ (*p* = 0.893), HR (*p* = 0.079), or labor effort (*p* = 0.357) (Figure 7). The “ice slurry” strategy did not impair workers’ T_core_ (*p* = 0.649), HR (*p* = 0.157), or labor effort (*p* = 0.118), but increased T_sk_ (*p* = 0.034; d = 0.94) compared to BAU (Figure 8).

#### 3.2.3. Tourism Industry

The tested heat mitigation strategies in the Greek tourism industry did not impact workers’ T_core_ (planned breaks: *p* = 0.430; ice slurry: *p* = 0.094; and combined: *p* = 0.135), T_sk_ (planned breaks: *p* = 0.909; ice slurry: *p* = 0.628; and combined: *p* = 0.326), HR (planned breaks: *p* = 0.384; ice slurry: *p* = 0.491; and combined: *p* = 0.536), or labor effort (planned breaks: *p* = 0.170; ice slurry: *p* = 0.992; and combined: *p* = 0.423) (Figure 9). Despite the lack of statistically significant differences based on *p* values, it is important to note that we found large effect sizes when comparing the T_core_ of workers between the BAU and either the “ice slurry” (d = 0.83) or the “combined” (d = 0.89) strategies, indicating that a larger sample size might have revealed a statistically significant difference.

## 4. Discussion

The present paper with observational and intervention studies showed that T_sk_ is positively associated with the OH-stress experienced by workers in agriculture, construction, and tourism industries. On the other hand, we found that workers’ T_core_ is associated with OH-stress only when self-pacing is not feasible or is very limited. For instance, the monitored construction and tourism workers had limited capacity for self-pacing which predisposed them to OH-strain. On the contrary, the studied agriculture workers maintain their T_core_ within a comfortable range by regulating their work intensity and consequently protecting themselves from experiencing increased OH-strain. Although these findings of limited self-pacing in construction and tourism workers do not necessarily reflect the work practices in the entire construction and tourism sectors, our results are in line with previous studies confirming that self-pacing can maintain workers’ T_core_ within a comfortable range despite being exposed to fluctuating harsh environmental conditions [38,39]. It is important to note that our findings of increased self-pacing in the agricultural sector may also reflect the higher metabolic demands characterizing the tasks performed by agriculture workers in comparison to people who work in the construction and tourism industries. Nevertheless, the observed increase in the T_core_ of workers who had limited ability for self-pacing highlights the proactive role of this important behavioral mechanism in the fight against OH-strain [40].

Although self-pacing is undoubtedly an important aspect of behavioral thermoregulation acting as a catalyst to reduce the OH-strain experienced by workers, it cannot guarantee adequate protection against OH-stress when used alone. A previous laboratory study [41] showed that self-determined breaks are too short to allow complete thermal recovery, and therefore it seems unwise to rely completely on self-pacing to avoid OH-strain even when workers are encouraged to adapt their own work pace [42]. It is more likely that the influence of exogenous environmental factors on the workers’ health and physiological function requires the adoption of complementary heat mitigation strategies. Indeed, here we show that several heat mitigation strategies can be adopted to provide workers with protective mechanisms that are both inexpensive to use and flexible in time schedule.

Reducing the metabolic heat production in workplaces with high OH-stress is of great importance, and mechanization is considered a fundamental approach to achieve that goal [43]. To test this hypothesis, we provided agriculture workers with a mechanical fruit cart able to carry up to 225 kg of crops with minimal labor effort. Although we did not find significant differences in the OH-strain experienced by agriculture workers when using the mechanical fruit cart compared to the BAU scenario, we showed that the use of the mechanical fruit cart led the workers to pick 63% more crops. Therefore, it is logical to assume that if a constant amount of crop was predefined, mechanization would be able to reduce the OH-strain experienced by agriculture workers. This was confirmed using the predicted heat strain model [30,44,45] which showed that the T_core_ of these agriculture workers would be 0.3 °C lower after carrying 225 kg of crops with the mechanical fruit cart compared to the BAU scenario (both scenarios were simulated for the same workers performing manual labor in the same environmental conditions, but having a different metabolic rate).

Providing workers with planned breaks is another widely accepted method to reduce their metabolic heat production and protect them against OH-strain. Several studies [46,47] previously examined the effects of the Threshold Limit Values [48] on the OH-strain experienced by workers. These work-rest cycles are based on WBGT and take into account acclimatization status and work intensity. However, although these work-rest cycles are adopted by many organizations including the US Occupational Safety and Health Administration [49], it has been shown that their applicability is limited in many occupational settings [46]. This, of course, is even more relevant in customer-depended industries and/or worksites where time can affect the quality of the final product or service. For this reason, we tested three different work-rest cycles tailored to our employers’ needs and requirements. Our findings show that providing workers with 90 s of break every half hour of continuous work did not significantly affect the OH-strain experienced by agriculture and tourism workers. Similarly, providing construction workers with two breaks of seven minutes throughout their work-shift did not alter their OH-strain. On the other hand, providing construction workers with ten minutes of break every 50 min of continuous work was able to reduce their T_sk_ and HR, while it did not affect their T_core_. Interestingly, the same heat mitigation strategy in agriculture showed a significant increase in the HR of workers, without affecting their T_sk_ and T_core_. This is likely due to changes in the body posture of workers from crouching (i.e., typical posture during crop picking) to standing and walking during breaks. These findings are in line with previous studies showing that short work-rest cycles are not beneficial in mitigating the heat strain experienced by individuals during physical work in the heat [21,47,50,51].

Hydration status is of vital importance for workers’ health and well-being, especially during work in the heat [1,6]. It is well established that hypohydration increases the heat stored in the body by limiting its ability to sweat and to send blood near the skin surface [52,53,54]. Our findings show that provision of 750 mL of water every hour can reduce the risk of dehydration by 54% among Qatar agriculture workers, by 97% among Qatar construction workers, and by 13% among Spanish construction workers. The hydration strategy also resulted in significantly reduced T_core_, and it did not impact labor effort. These findings support the notion that hydration is the single most important and economically feasible mitigation strategy against OH-strain.

Ice slurry consumption is another strategy previously used to mitigate the heat strain experienced by athletes [55,56] and workers [24] who perform physical work in the heat. It is considered a highly effective strategy that takes advantage of the large (334 kJ/kg) latent heat of fusion of water (quantity of energy required to change the phase of water from solid to liquid). Due to its limited practicality for remote work sites, we tested this strategy in construction and tourism. Our results showed a large effect size on T_core_ in Greek tourism workers, but no statistically significant differences on T_core_ in Greek tourism or Spanish construction workers. These findings are in line with a previous study [56] reporting no significant differences in the T_core_ and T_sk_ of individuals who perform exercise in the heat between ice slurry ingestion and 37 °C fluid ingestion. Therefore, it is reasonable to suggest that the benefits of ice slurry ingestion in our study were diluted by the cooling impact of other fluids that the workers consumed during the BAU scenario.

Clothing is ranked among the most important exogenous parameters affecting the efficiency of evaporative, convective, and radiative heat losses, leading to intense body heat gain in the heat [57]. Although this is a very well-known fact, there are technical aspects (e.g., need for protective clothing, helmet, boots, dress code, etc.) that may restrict the adoption of more efficient clothing strategies in occupational settings. Taking into account the aforementioned restrictions, we examined the capacity of two different garments to mitigate the OH-strain experienced by workers in construction and agriculture industries. Our results indicate that providing agriculture workers with ventilated garments (short-sleeved shirts with integrated electric fans) reduced considerably their T_sk_ but did not affect their T_core_, HR, or labor productivity during the work-shift. This finding is in line with previous studies showing that reducing physiological strain (i.e., T_sk_) can be accomplished by the use of garments which provide greater ventilation and consequently enhanced evaporative and convective heat losses, even though T_core_ was not altered significantly [58]. On the other hand, providing workers with evaporative garments (breathable shirt and trousers allowing the wearer to wet parts of the garment) did not significantly affect workers’ OH-strain or labor effort in construction or agriculture sectors. Observations made by our research team suggested that many workers did not follow the manufacturer’s usage instructions for frequent wetting and often wore t-shirts underneath the evaporative garments.

It is important to note that the present study involved monitoring people working in developed countries, and thus our results may not reflect the practices and conditions under which workers perform their jobs in other parts of the world. Furthermore, although this series of field experiments involved a comparatively large sample size across different industries and countries, more focused interventions for gender-, age-, and body- specific differences should be conducted to elucidate the capacity of different interventions to mitigate the physiological heat strain experience by different working populations. Additionally, the rapid technological progress that is currently taking place is expected to enable new capabilities soon regarding the variety of physiological indicators that ecological studies can measure. For instance, wireless sweat rate and skin blood flow data loggers could soon be utilized to uncover even further aspects of occupational heat strain, triggering the development of more focused heat mitigation strategies. Furthermore, this technological progress may also enable the development of new techniques to examine labor productivity, minimizing possible differences between the time-motion analysis and real-time task analysis that were used in the present study. Another important limitation of our study is that the tested heat mitigation strategies were examined in various environmental conditions, and therefore we cannot directly compare the findings observed across different industries. For instance, agricultural tasks are characterized by prolonged exposure to intense solar radiation which may affect workers’ capacity to perform their jobs, compared to people who work in the construction and tourism sectors. It is also important to note that, during the BAU scenario in the construction sectors of Qatar and Spain, a raft of other heat mitigation strategies was already in place, and this may not reflect the “business as usual” in other countries/workplaces. Furthermore, although our study involves testing different heat mitigation strategies across multiple countries characterized by high heat stress, we found that the monitored workers experienced low-to-moderate OH-strain, and therefore this may undermine the capacity of a heat mitigation strategy to reduce the OH-strain experienced by someone. Another important factor which may undermine the capacity of the tested heat mitigations strategies is self-pacing, indicating that in workplaces where self-pacing is limited, heat mitigation strategies may be of higher importance.

## 5. Conclusions

Taken together, our findings show that workers in the agriculture, construction, and tourism industries across Cyprus, Greece, Qatar, and Spain experience high levels of OH-stress accompanied by low-to-moderate OH-strain. However, when self-pacing is not feasible or very limited, the OH-stress experienced by workers leads to higher levels of OH-strain. Strategies targeting hydration, work-rest cycles, and ventilated garments show the most promising results towards mitigating OH-strain. Moreover, increasing mechanization—particularly for the most physically demanding tasks—can enhance labor productivity without increasing OH-strain.

## Figures and Tables

**Figure 1 ijerph-18-06303-f001:**
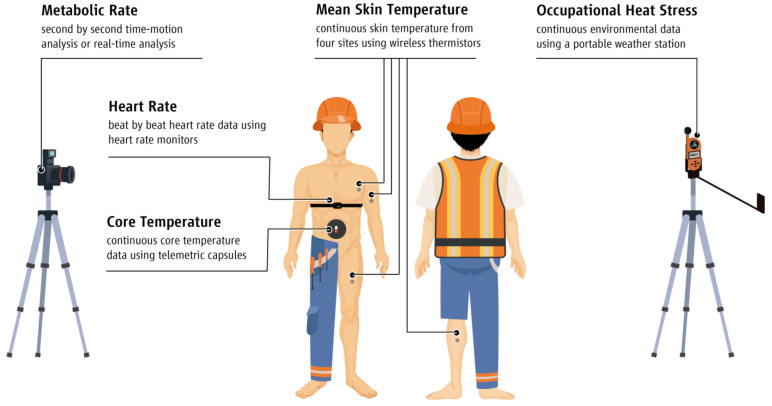
Data collection throughout the experiments.

**Figure 2 ijerph-18-06303-f002:**
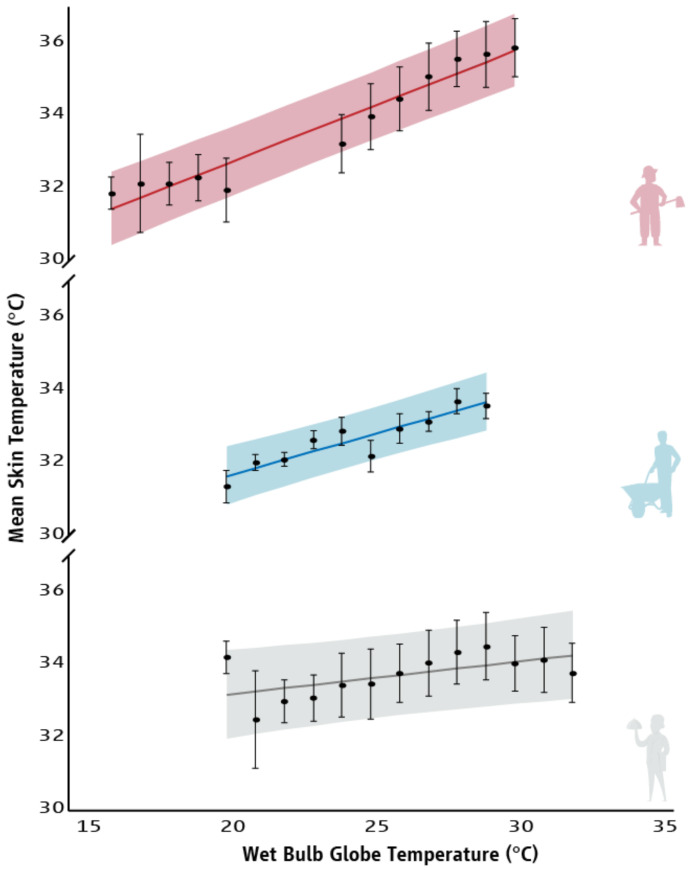
Relationship between Wet-Bulb Globe Temperature and the mean skin temperature (average ± SD) of people who work in agriculture (red), construction (blue), and tourism (grey) sectors. Lines represent the slope of the predicted relationship between Wet-Bulb Globe Temperature and the mean skin temperature of workers. Shadings correspond to the 95% prediction intervals of the means.

**Figure 3 ijerph-18-06303-f003:**
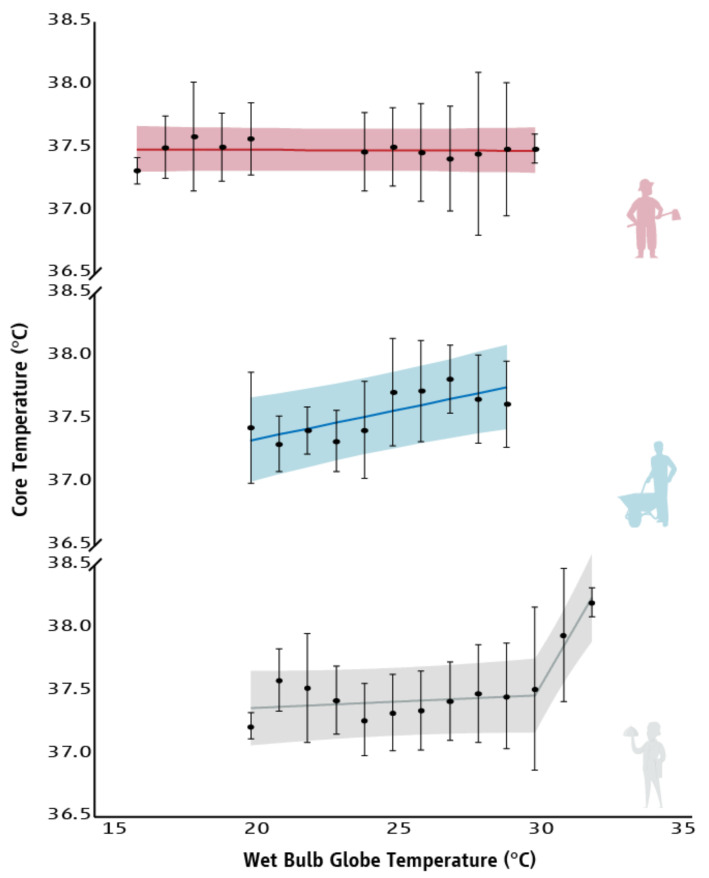
Relationship between Wet-Bulb Globe Temperature and the core temperature (average ± SD) of people who work in agriculture (red), construction (blue), and tourism (grey) sectors. Lines represent the slope of the predicted relationship between Wet-Bulb Globe Temperature and the core temperature of workers. Shadings correspond to the 95% prediction intervals of the means.

**Figure 4 ijerph-18-06303-f004:**
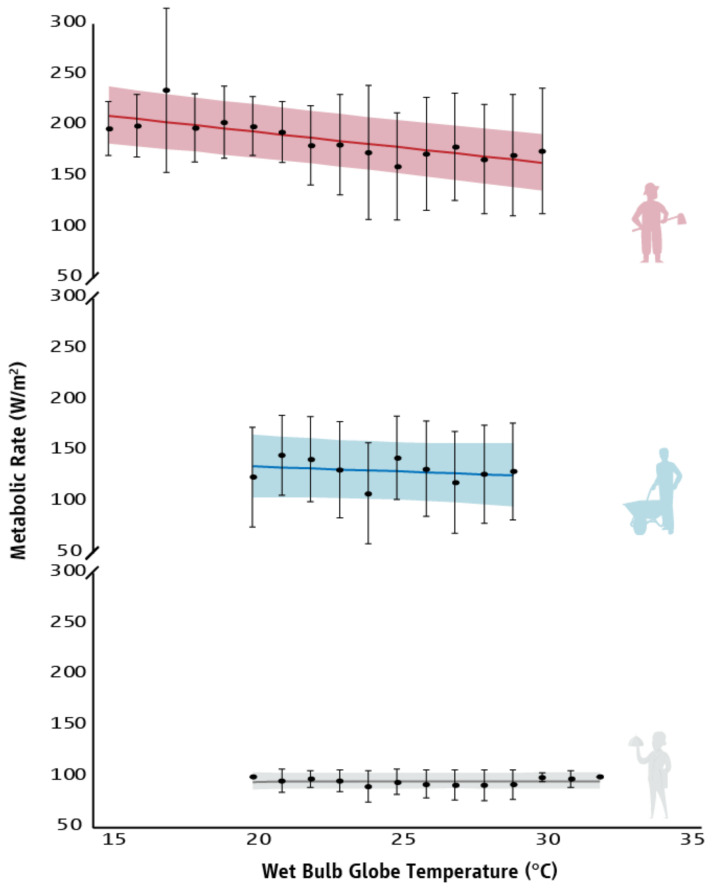
Relationship between Wet-Bulb Globe Temperature and the metabolic rate/work intensity (average ± SD) of people who work in agriculture (red), construction (blue), and tourism (grey) sectors. Lines represent the slope of the predicted relationship between Wet-Bulb Globe Temperature and the metabolic rate/work intensity of workers. Shadings correspond to the 95% prediction intervals of the means.

**Figure 5 ijerph-18-06303-f005:**
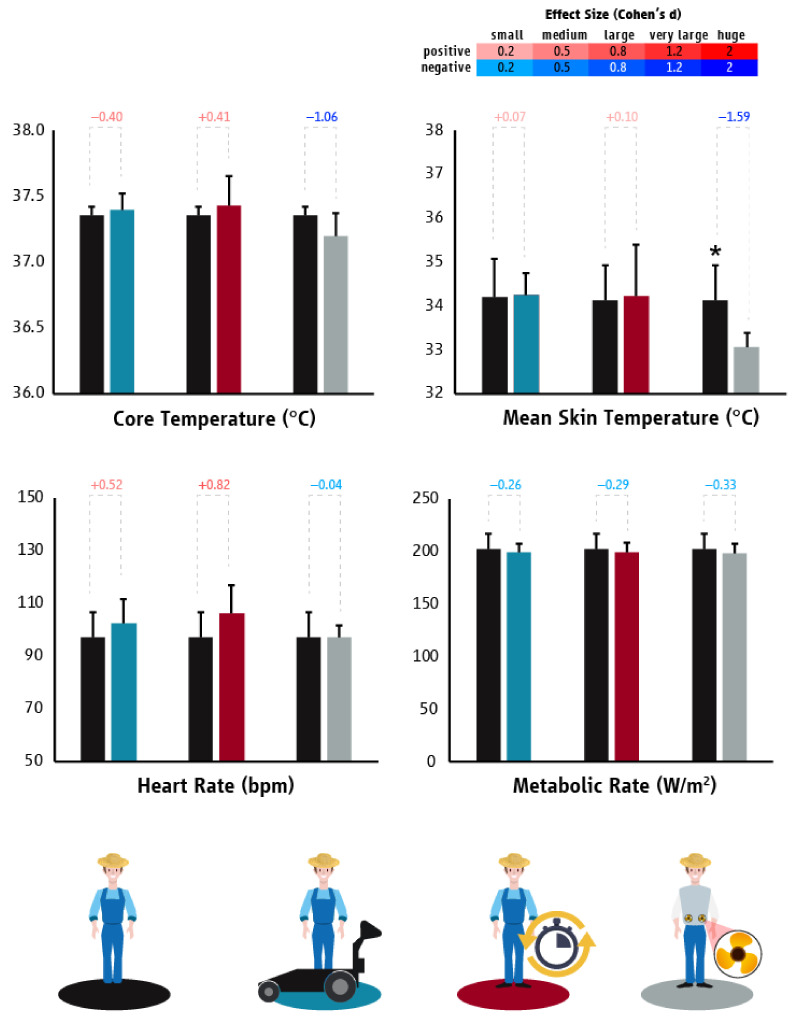
Differences (mean ± SD) in core temperature, mean skin temperature, heart rate, and metabolic rate/work intensity between “business as usual” and the tested heat mitigation strategies in the agriculture sector of Cyprus. Black, light blue, red, and grey colors represent “business as usual”, mechanical fruit cart, planned breaks, and ventilated garments scenarios, respectively. Asterisk indicates statistically significant difference between “business as usual” and the tested heat mitigation strategy. Cohen’s d effect sizes show the magnitude (small: 0.2; medium: 0.5; large: 0.8; very large: 1.2; huge: 2.0) and direction (positive: shades of red; negative: shades of blue) of the differences between “business as usual” and the tested heat mitigation strategies.

**Figure 6 ijerph-18-06303-f006:**
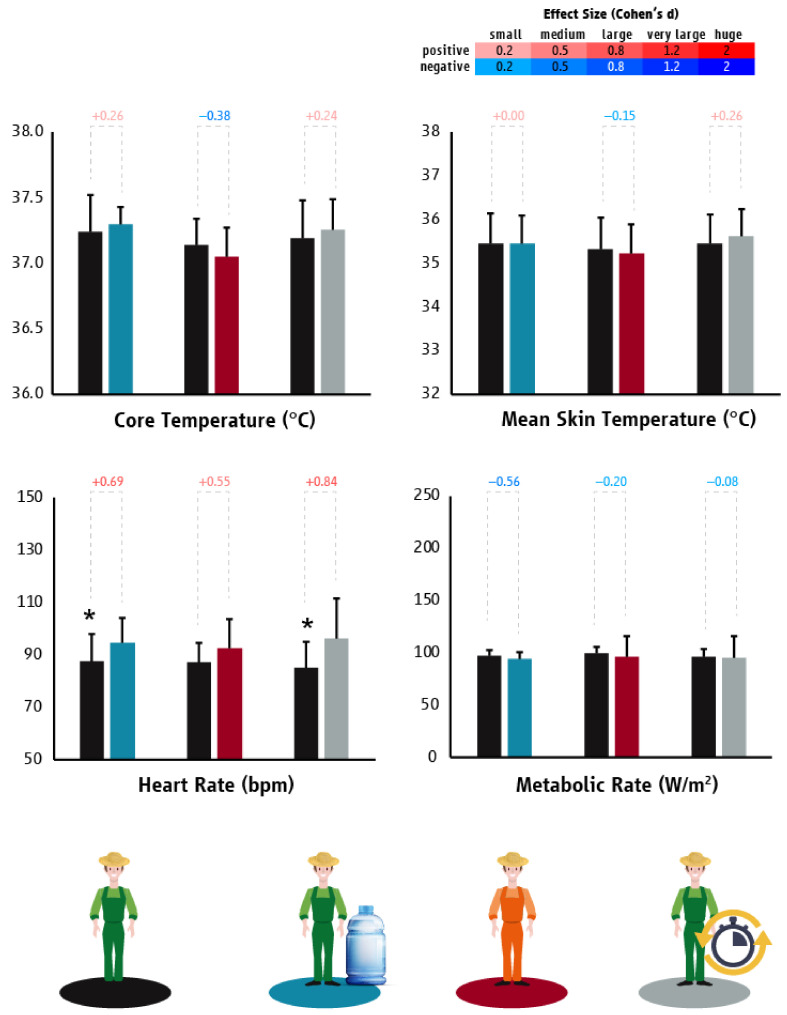
Differences (mean ± SD) in core temperature, mean skin temperature, heart rate, and metabolic rate/work intensity between “business as usual” and the tested heat mitigation strategies in the agriculture sector of Qatar. Black, light blue, red, and grey colors represent “business as usual”, hydration, evaporative garments, and planned breaks scenarios, respectively. Asterisks indicate statistically significant differences between “business as usual” and the tested heat mitigation strategies. Cohen’s d effect sizes show the magnitude (small: 0.2; medium: 0.5; large: 0.8; very large: 1.2; huge: 2.0) and direction (positive: shades of red; negative: shades of blue) of the differences between “business as usual” and the tested heat mitigation strategies.

**Figure 7 ijerph-18-06303-f007:**
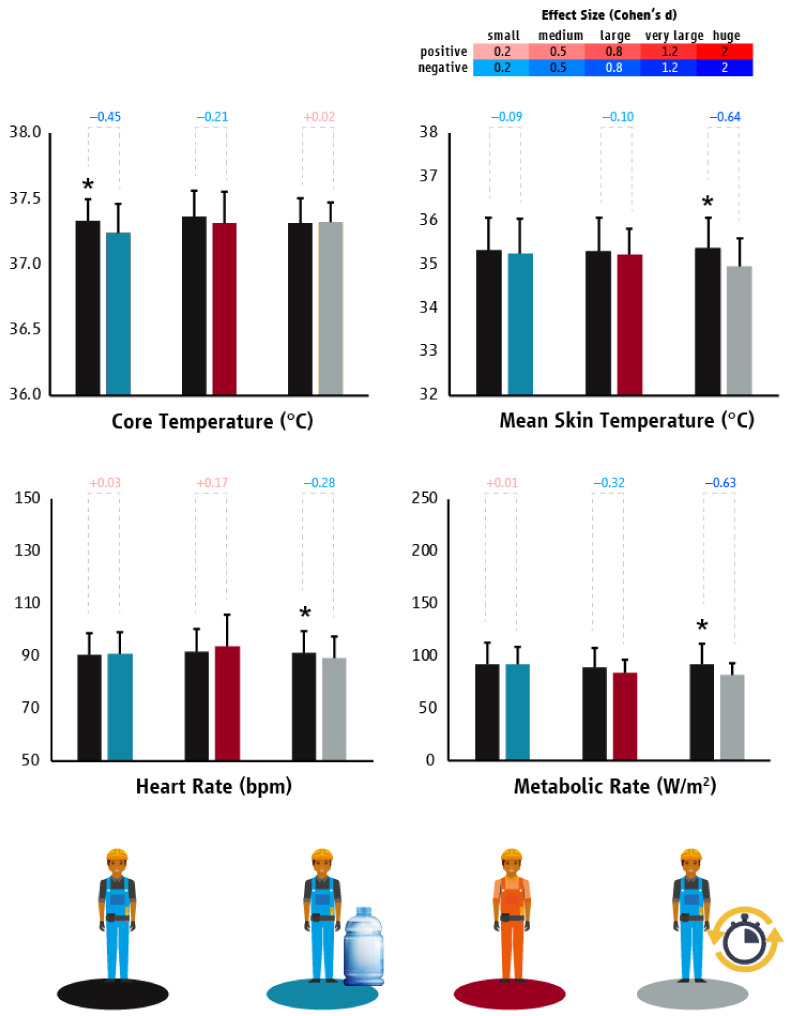
Differences (mean ± SD) in core temperature, mean skin temperature, heart rate, and metabolic rate/work intensity between “business as usual” and the tested heat mitigation strategies in the construction sector of Qatar. Black, light blue, red, and grey colors represent “business as usual”, hydration, evaporative garments, and planned breaks scenarios, respectively. Asterisks indicate statistically significant differences between “business as usual” and the tested heat mitigation strategies. Cohen’s d effect sizes show the magnitude (small: 0.2; medium: 0.5; large: 0.8; very large: 1.2; huge: 2.0) and direction (positive: shades of red; negative: shades of blue) of the differences between “business as usual” and the tested heat mitigation strategies.

**Figure 8 ijerph-18-06303-f008:**
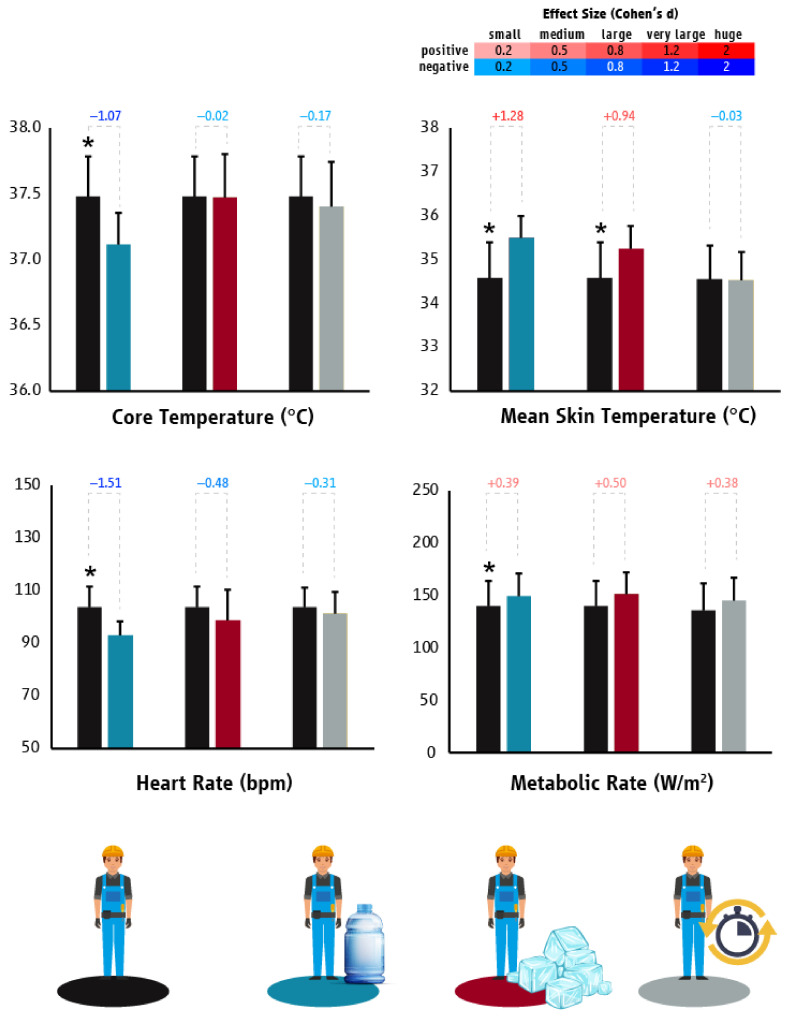
Differences (mean ± SD) in core temperature, mean skin temperature, heart rate, and metabolic rate/work intensity between “business as usual” and the tested heat mitigation strategies in the construction sector of Spain. Black, light blue, red, and grey colors represent “business as usual”, hydration, ice slurry, and planned breaks scenarios, respectively. Asterisks indicate statistically significant differences between “business as usual” and the tested heat mitigation strategies. Cohen’s d effect sizes show the magnitude (small: 0.2; medium: 0.5; large: 0.8; very large: 1.2; huge: 2.0) and direction (positive: shades of red; negative: shades of blue) of the differences between “business as usual” and the tested heat mitigation strategies.

**Figure 9 ijerph-18-06303-f009:**
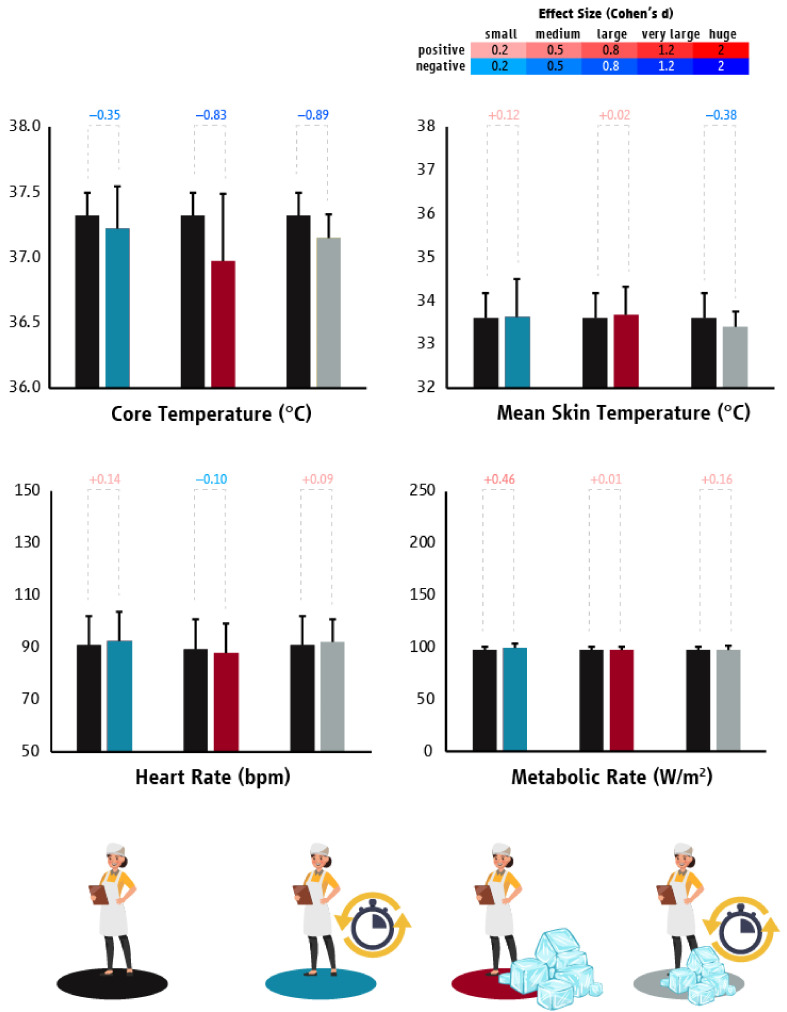
Differences (mean ± SD) in core temperature, mean skin temperature, heart rate, and metabolic rate/work intensity between “business as usual” and the tested heat mitigation strategies in the tourism sector of Greece. Black, light blue, red, and grey colors represent “business as usual”, planned breaks, ice slurry, and “combined” (two minutes of planned break combined with ice slurry consumption (2.4 g per body mass kilogram) every hour of continuous work) scenarios, respectively. Cohen’s d effect sizes show the magnitude (small: 0.2; medium: 0.5; large: 0.8; very large: 1.2; huge: 2.0) and direction (positive: shades of red; negative: shades of blue) of the differences between “business as usual” and the tested heat mitigation strategies.

**Table 1 ijerph-18-06303-t001:** Workers’ personal characteristics (mean ± SD). Asterisk indicates that no anthropometric data were collected from all participants.

**Observational Studies**				
Sector	Workers (*n*)	Weight (kg)	Height (cm)	Age (Years)
Agriculture (Greece)	36 (7) *	75.4 ± 13.2	169.1 ± 4.9	39.9 ± 14.2
Construction (Spain)	14	79.6 ± 11.1	174.3 ± 8.9	43.3 ± 10.2
Tourism (Greece)	49	75.6 ± 14.7	1.7 ± 0.1	34.5 ± 9.5
**Interventional Studies**				
	Workers (*n*)	Weight (kg)	Height (cm)	Age (years)
Agriculture (Cyprus) *n* = 6				
Work/rest ratio	6	77.0 ± 16.2	168.3 ± 8.5	39.2 ± 11.8
Fruit cart (eCart)	6	77.0 ± 16.2	168.3 ± 8.5	39.2 ± 11.8
Ventilated garments	6	77.0 ± 16.2	168.3 ± 8.5	39.2 ± 11.8
Agriculture (Qatar) *n* = 34				
Work/rest ratio	24	66.4 ± 10.0	170.6 ± 5.7	31.5 ± 7.5
Hydration	26	65.2 ± 9.4	169.7 ± 5.8	32.1 ± 7.2
Evaporative garments	12	67.3 ± 9.2	170.1 ± 4.3	35.3 ± 8.5
Construction (Qatar) *n* = 83				
Work/rest ratio	69	65.3 ± 8.4	164.9 ± 5.7	34.4 ± 8.3
Hydration	53	65.7 ± 8.1	165.2 ± 5.8	34.3 ± 9.2
Evaporative garments	32	65.1 ± 8.1	164.5 ± 5.9	35.9 ± 7.8
Construction (Spain) *n* = 10				
Work/rest ratio	10	85.9 ± 14.4	175.8 ± 10.9	41.5 ± 7.3
Hydration	9	77.9 ± 15.0	158.2 ± 11.5	39.0 ± 5.0
Ice slurry	9	77.9 ± 15.0	158.2 ± 11.5	39.0 ± 5.0
Tourism (Greece) *n* = 6				
Work/rest ratio	6	71.2 ± 9.8	171.2 ± 7.4	30.5 ± 8.3
Ice slurry	6	71.2 ± 9.8	171.2 ± 7.4	30.5 ± 8.3
Combined	6	71.2 ± 9.8	171.2 ± 7.4	30.5 ± 8.3

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
