# Peer review of "Occupational Heat Stress: Multi-Country Observations and Interventions"

_ijerph, 2021, doi:10.3390/ijerph18126303_

Round 1
Reviewer 1 Report
First of all, I would like to thank you for having me as a reviewer of this publication. Without a doubt, I consider that the article brings great value to the business world and, thanks to the information provided, healthy policies can be designed and implemented to reverse this situation of workers.
Abstract:
22-23: try not to use the pronoun "we". Impersonal or passive voice should be used.
Introduction:
I consider it to have a common thread. It is well laid out, puts the reader in context and allows the reader to understand the current situation and the real problem of the issue.
Materials and Methods
Line 93-95: Consider omitting the description of the objective or formulating it differently.
Consider including a picture, diagram, figure where all the variables are explained. It is confusing for the reader. Or rewrite more neatly.
169: Urine errata (omit space)
176: only 90 seconds of break?
Indicate that there is a summary table (1) below.
I didn't see inclusion or exclusion criteria, have you used them?
Results:
The results are well laid out. They are well understood, and the graphs are very dynamic. It helps understanding.
Discussion:
It is well stated, it refers to all the results raised in the previous section.
Conclusions:
I have not seen any limitations of the study, I consider it important to include them.
Reviewer 2 Report
See attached pdf file for comments and suggestions to authors.

Reviewer 3 Report
An interesting and comprehensive study of the effect of heat on workers although the paper pushes the quite limited results to limits that I don't think warrant the research.
The abstract and Introduction focus the reader on health and mortality issues while the research is primarily on productivity loss.
For example on line 44 " Globally, three out of ten people are currently exposed to environmental conditions exceeding deadly climatic thresholds for at least twenty days per 45 year." The quoted research is not about working people but about elderly people and I think it unfair (and confusing) to mix up death from classic heat stroke with that from exertional heat stroke. Most of the increased mortality is not in working people. This early focus in the introduction (and the abstract needs to be either toned down or removed.
The other major issue I have is around figure 2.
The error bars are large and the 95% confidence interval of the predicted slope is incorrectly shown. A quick back of the envelop calculation shows that to a 95% level of confidence 0 slope falls in that range for the middle graph. Indeed, if the shaded area shows a 95% confidence area for slope, then the second graph (construction) as a slope = 0 within that 95% confidence interval - ie from the TOP left corner of the shaded region to the BOTTOM right corner of the shaded region. As for the third graph (tourism) it is very clear from the graph that there is ZERO slope from 20 WBGT to 30 WBGT and then it takes off. This is very much in line with ISO/NIOSH data and Sahu/Wyndham data that shows no productivity loss till about 30 WBGT and then it increases rapidly. To say the slope starts to rise at 20 WBGT is misleading.
Other more minor matters:
Lines 123-154 Contain much repetitive detail that would go much better into a table.
Also metrics units should be used throughout (lbs line 140 and cal/g on line 489.
Line 159 BAU is a term often used with climate change in relation to no reduction of GHG. Maybe Working As Usual (WAU) would be a better acronym.
Under observational studies a number of types of equipment are specified, but there is no mention of a WBGT meter. Was the WBGT derived from temperature and humidity or was a meter used - and what type of meter?
On line 245 a windspeed of 0.5m/s is used. It seems very low. Did you take the apparent windspeed on the workers moving quickly through the fields into account when determining WBGT?
Figures 6 and 7 need to be checked for * significance. Evaporative Garments under heart rate dont appear significant and Metabolic Rate under hydration and ice have almost identical heights and error bars and one is significant and the other is not.
Line 425. It is NOT correct that agricultural workers do not have a time line they have to meet (like the coming of rain in harvest season or the milk truck arriving early in the morning to pick up milk). While it might be true that this was the case for the agricultural workers in this study, it is not universally true.
Line 266 to 270. It is unrealistic to compare the benefit a 90 second break with a 10 minute break.
Table A1 is pretty useless because the SD is so high and the mean + SD of the items does not match in relation to time. Also a summary was in the main text. It would be MUCH more useful for the actual anonymous data to be supplied in a spreadsheet than these means and SDs.
Reference 10 is very dated (and is grey literature) and has been updated since: Kjellstrom et al 2018 (Int J Biomet)
Reviewer 4 Report
The manuscript addresses an important and relevant issue regarding the occupational heat stress. The study involved 518 full work-shifts obtained from 238 experienced and acclimatized workers. The objectives are stated clearly, but methods are unclear: there is no information about the criteria adopted for selecting the 238 workers. In particular, I think that the results of the study could be distorted by bias. The aforementioned issue represents a “selection bias” and does not allow to attribute significance to the results of the study. Given this issue, the authors should revise the methods and exclude the risk of selection bias. Limitations of the study must be stated.
Round 2
Reviewer 2 Report
See attached file.

Reviewer 3 Report
It is good that some of the minor comments I pointed out have been corrected. But there are still some major comments I made on the forst paper that have not been adequately addressed.
It may have been that I was not completely clear of my concerns in my first version of comments. This time I have elaborated on them and included a diagram in the attached file.

Reviewer 4 Report
The authors answered my questions.
Author Response
Thank you very much for your comments which helped us to improve the overall quality of our paper.
Round 3
Reviewer 2 Report
I have no further comment